# Invasive Diagnostic Procedures from Bronchoscopy to Surgical Biopsy—Optimization of Non-Small Cell Lung Cancer Samples for Molecular Testing

**DOI:** 10.3390/medicina59101723

**Published:** 2023-09-27

**Authors:** Nensi Lalić, Aleksandra Lovrenski, Miroslav Ilić, Olivera Ivanov, Marko Bojović, Ivica Lalić, Spasoje Popević, Mihailo Stjepanović, Nataša Janjić

**Affiliations:** 1Faculty of Medicine in Novi Sad, University of Novi Sad, Hajduk Veljkova 3, 21137 Novi Sad, Serbia; aleksandra.lovrenski@mf.uns.ac.rs (A.L.); mirslav.ilic@mf.uns.ac.rs (M.I.); olivera.ivanov@mf.uns.ac.rs (O.I.); marko.bojovic@mf.uns.ac.rs (M.B.); natasa.janjic@mf.uns.ac.rs (N.J.); 2Institute for Pulmonary Diseases of Vojvodina, 21204 Sremska Kamenica, Serbia; 3Clinic of Radiation Oncology, Oncology Institute of Vojvodina, 21204 Sremska Kamenica, Serbia; 4Faculty of Pharmacy, University Business Academy in Novi Sad, Trg Mladenaca 5, 21101 Novi Sad, Serbia; ivica.lalic@faculty-pharmacy.com; 5Faculty of Medicine, University of Belgrade, 11000 Belgrade, Serbia; spasoje.popevic@med.bg.ac.rs (S.P.); mihailo.stjepanovic@med.bg.ac.rs (M.S.); 6University Hospital of Pulmonology, Clinical Center of Serbia, 11000 Belgrade, Serbia

**Keywords:** advanced lung cancer, bronchoscopy, liquid biopsies, rebiopsy, tumor molecular biology

## Abstract

*Background and Objectives*: Treatment of advanced lung cancer (LC) has become increasingly personalized over the past decade due to an improved understanding of tumor molecular biology and antitumor immunity. The main task of a pulmonologist oncologist is to establish a tumor diagnosis and, ideally, to confirm the stage of the disease with the least invasive technique possible. *Materials and Methods*: The paper will summarize published reviews and original papers, as well as published clinical studies and case reports, which studied the role and compared the methods of invasive pulmonology diagnostics to obtain adequate tumor tissue samples for molecular analysis, thereby determining the most effective molecular treatments. *Results*: Bronchoscopy is often recommended as the initial diagnostic procedure for LC. If the tumor is endoscopically visible, the biopsy sample is susceptible to molecular testing, the same as tumor tissue samples obtained from surgical resection and mediastinoscopy. The use of new sampling methods, such as cryobiopsy for peripheral tumor lesions or cytoblock obtained by ultrasound-guided transbronchial needle aspiration (TBNA), enables obtaining adequate small biopsies and cytological samples for molecular testing, which have until recently been considered unsuitable for this type of analysis. During LC patients’ treatment, resistance occurs due to changes in the mutational tumor status or pathohistological tumor type. Therefore, the repeated taking of liquid biopsies for molecular analysis or rebiopsy of tumor tissue for new pathohistological and molecular profiling has recently been mandated. *Conclusions*: In thoracic oncology, preference should be given to the least invasive diagnostic procedure providing a sample for histology rather than for cytology. However, there is increasing evidence that, when properly processed, cytology samples can be sufficient for both the cancer diagnosis and molecular analyses. A good knowledge of diagnostic procedures is essential for LC diagnosing and treatment in the personalized therapy era.

## 1. Introduction

At diagnosis, 75% of LC patients have locally advanced or metastatic disease. Treatment of advanced LC has become increasingly personalized over the past decade due to an improved understanding of tumor molecular biology and antitumor immunity. The main task of a thoracic oncologist is to establish the diagnosis and, ideally, to confirm the stage of the disease with the least invasive technique possible. As a result of this approach, biopsy samples are becoming smaller and smaller. Advanced diagnostic bronchoscopy techniques play a central role in evaluating patients with LC. An adequate tumor sample is crucial for targeted mutational analysis and immune profiling. When asked: “Are we getting enough?”, the answer is “It depends”—mainly on the procedure being performed, but also partly on the required analysis!

The treatment of LC has evolved far beyond a simple therapeutic option based on the histological difference between small cell lung cancer (SCLC) and non-small cell lung cancer (NSCLC). New treatment modalities for NSCLC, such as targeted therapy, have significant advantages over standard cytotoxic chemotherapy. Patients with epidermal growth factor receptor (EGFR) mutations, typically in exons 19 and 21, show dramatic responses to specific tyrosine kinase inhibitors (TKIs) such as gefitinib, erlotinib, afatinib, and osimertinib (10–15% of NSCLC) [1,2]. The presence of the anaplastic lymphoma kinase (ALK) gene rearrangement responds to treatment with ALK inhibitors such as crizotinib, ceritinib, alectinib, and brigatinib [3]. ROS-1 rearrangement (1–2% of NSCLC) also predicts response to crizotinib [4]. KRAS mutation, detected mainly in smokers with lung adenocarcinoma, is associated with a poor disease prognosis and poor response to systemic and molecular therapy [5]. Rapidly advancing immunotherapy produced benefits for patients with LC, as well as a clinical improvement for patients with KRAS mutation NSCLC [6,7]. Over the past decade, the understanding of antitumor immunity has dramatically improved. Interactions of programmed death-ligand 1 (PD-L1) with its receptor are recognized as an essential immune mechanism by which tumor cells avoid the body’s defensive immune response. The basis of immunotherapy in patients with LC is inhibiting the tumor PD-L1 expression. The level of this expression is measured by the immunohistochemical analysis, whereby patients with advanced stage NSCLC and PD-L1 expression of at least 50% of tumor cells (tumor proportion score or TPS 50%) have benefited from the therapy with anti-PD-1 and anti-PD-L1 immunotherapeutics in terms of survival and reduction of adverse events compared to standard chemotherapy. In patients with a lower PD-L and PD-L1 expression score of 50% (1–49%), immunotherapy should be combined with chemotherapy, while patients with an expression of 1% may benefit from immunotherapy in the second line of treatment [8]. Small biopsies and cytology samples may increase the risk of false-negative PD-L1 results based on its tumor expression’s known temporal and spatial heterogeneity. Advanced molecular testing that identifies targeted mutations and quantifies PD-L1 expression leads to personalized therapy for patients with LC. Most patients with advanced LC are diagnosed with minimally invasive techniques, which yield small biopsies and cytological samples that must be precisely processed for molecular testing. Small biopsies and cytological samples also play a central role in selecting new therapeutic options in disease progression after initial treatment [9].

The latest guidelines of the College of American Pathologists (CAP), the International Association for the Study of Lung Cancer (IASLC), and the Association for Molecular Pathology (AMP) suggest molecular tests for the initial diagnosis of lung adenocarcinoma in an advanced stage of the disease [10]. The National Comprehensive Cancer Network (NCCN) has expanded the indications for molecular testing for EGFR and ALK to other types of NSCLC, not otherwise specified (NOS) NSCLC, as well as to patients with metastatic disease and squamous LC who are non-smokers [11]. In subsequent versions of the NCCN guidelines, the panel of molecular tests was expanded to include BRAF, ROS-1, RET, ERBB2 (HER2), KRAS, and MET mutations as routine for patients with lung adenocarcinoma [12].

The CAP/IASLC/AMP guidelines suggest different assays for mutation detection where the proportion of tumor cells in the sample must be at least 20%. The NCCN guide suggests polymerase chain reaction (PCR), fluorescence in situ hybridization (FISH), Sanger sequencing, and other multiple tests such as SNaPshot and MassARRAY, but the next-generation sequencing (NGS) test is considered the latest and most competent analysis of all. NGS enables the detection of a large number of mutations at the same time when all other tests are inadequate in terms of deficient tissue samples [13,14].

## 2. Optimizing Tissue Sampling for LC Diagnosis, Subtyping and Molecular Analysis

LC guidelines recommend prompt diagnosis and referral for treatment. The goal is to reduce the waiting time and start the diagnostic procedure based on established algorithms to provide the most information about the diagnosis and stage of the disease with the least risk for the patient. Bronchoscopy, with or without lymph node sampling, is often recommended as the initial diagnostic procedure for LC. Flexible bronchoscopy is usually performed with local anesthesia and minimal sedation, ensuring visualization of all segmental bronchi. Complications are rare and amount to about 5%, most commonly including pneumothorax, bleeding, or hypoxia. An endobronchial tumor can be visualized as an exophytic mass or submucosal infiltration (Figure 1). When the endobronchial lesion is visible, diagnostic positivity is achieved in 70–90% [15]. It has been proven that five taken samples are the optimal number for reaching the diagnosis in central, endoscopically visible lesions, and the sensitivity increases when a brush, lavage, and catheter biopsy are concurrently performed [16]. Although five samples are considered sufficient for the diagnosis, the number of samples for a detailed subclassification and molecular tests has not been precisely determined.

### 2.1. Cryobiopsy for Endoscopy Visible Tumor Lesions and Transbronchial Cryobiopsy for Peripheral Tumor Lesions—Application of Molecular Tests

Cryobiopsy, as a recent method in diagnostic bronchoscopy (it also has a therapeutic function), has improved the diagnostic sensitivity of biopsy materials, providing larger samples for both central and peripheral lung lesions. Cryobiopsy enables obtaining endobronchial tumor tissue by the principle of “freezing” and extracting the tumor tissue attached to the tip of the cryoprobe itself. Several studies have compared the diagnostic sensitivity using cryobiopsy or conventional forceps biopsy in endobronchial lesions. In their work, which included 600 patients in eight bronchology centers [18], Hetzel et al. determined that the sensitivity of endobronchial forceps biopsy was 85.1%, and that of cryobiopsies was 95% (*p* < 0.001), thus proving the superiority of cryobiopsy as a diagnostic procedure for patients with LC [19]. Rubio et al. obtained a 96.77% sensitivity of cryobiopsy for LC. In their study, Kvale et al. documented the diagnostic sensitivities of cryoprobe and conventional forceps biopsy of 92% and 78%, respectively [20,21]. Transbronchial biopsy (TBB) was initially introduced as a diagnostic method for diffuse lung diseases and now is a relatively safe technique to obtain an adequate sample in patients with LC [22]. Diagnostic sensitivity is low (40%), compared to that in endoscopically visible tumor lesions. The sensitivity is undoubtedly increased when a bronchoscopic procedure is preceded by computed tomographic visualization. Indications for TBB are peripheral lung lesions, possibly with a positive bronchial sign seen by computed tomography, as well as the lesions with the characteristic “tree in the bud,” alveolar consolidation and reticulonodular changes in perilymphatic distribution. Transbronchial Lung Cryobiopsy (TBLC) is performed with a 1.9 to 2.4 mm working channel bronchoscope for diagnosing peripheral lung lesions. It is used with the Forgati catheter, placed in the segmental bronchus, and prevents severe bleeding after taking a cryobiopsy from the place of interest (Figure 2) [23,24]. In one of the studies, the sensitivity of TBLC using forceps biopsy or forceps biopsy followed by cryobiopsy guided by radial probe endobronchial ultrasound with a guided sheath (RP-EBUS-GS) has been shown. Diagnostic yields were compared between forceps biopsy and cryobiopsy, which amounted to 86.8% and 81.1% (OR = 11.6, *p* = 0.044) [25]. Imabayashi et al. obtained the diagnostic histological sensitivity of 86.1% in peripheral lung lesions using TBLC with EBUS-GS guidance before but not simultaneously with TBLC [26]. The quality of molecular analysis is presented in a large prospective study (single-armed study) involving a total of 121 patients with a suspected or diagnosed peripheral lung tumor. Diagnostic and molecular results were compared using TBLC or TBB performed with a flexible bronchoscope (1.60 mcg DNA and 0.62 mcg RNA cryoprobe—TBLC) vs. (0.58 mcg DNA and 0.17 mcg RNA TBB forceps with conventional bronchoscopy without cryoprobe). TBLC also surpassed TBB in obtaining PD-L1 expression >1% (51% vs. 42%) [27]. Another study also reported a successful NGS with DNA analysis that was achieved using TBLC, where 17 samples were sequenced, providing DNA of high quality and quantity [18].

### 2.2. Radial Probe—Endobronchial Ultrasound with a Guide Sheath in the Diagnosis of Peripheral Lung Lesions

Endobronchial ultrasonography with a guide sheath (EBUS-GS), which in bronchology is also termed “a radial probe” (RP-EBUS), increases the diagnostic sensitivity for peripheral lung lesions (PPLs). In PPLs, the diagnostic sensitivity of conventional bronchoscopy is 36%, compared to the 58–77% diagnostic sensitivity of radial EBUS. EBUS-GS can “differentiate” three types of peripheral lung lesions: type I (homogeneous opacity), type II (hyperechoic punctate or linear shadow shape), and type III (heterogeneous shadow shape) [28]. A meta-analysis showed that RP-EBUS has the specificity of 1.00 (95% confidence interval (CI), 0.99–1.00) and the sensitivity of 0.73 (95% CI, 0.70–0.76) in the diagnosis of LC. Sensitivity is higher for lesions larger than 2 cm (77.7%; 95% CI, 73–82%) than for lesions smaller than 2 cm (56.3%; 95% CI, 51–61%). The diagnostic value also depends on the position of the lesion in relation to the radial probe (Figure 3) [29].

If the probe is inserted within the lesion, the highest diagnostic sensitivity of 87% is obtained, compared to the maximum 42% sensitivity achieved by the probe inserted next to the lesion [30].

#### RP-EBUS-GS—Guided Endobronchial Ultrasound in the Era of Molecular Testing of Tumor Tissue

There are relatively few data on the quality and reliability of molecular testing on samples obtained by radial ultrasound [31]. Moon et al. reported the results of their research in which the adequacy of molecular testing on TBB tumor tissue samples, obtained by RP-EBUS-GS, was 63 of 64 samples (98%) adequate for EGFR assay, 60 of 60 (100%) adequate for ALK IHC, and 16 of 17 (94%) for PD-L1 IHC [32]. In their paper, Kim et al. analyzed the adequacy of the material for EGFR and ALK rearrangement by comparing the samples obtained by TBB—guidance by RP-EBUS and surgical resections. Adequacy was 97%/100% for EGFR, 100%/100% for ALK [33] in favor of the RP-EBUS. One of the studies obtained the adequacy of samples for molecular tests using RP-EBUS of 111 nonsquamous-NSCLC, 88 of the samples were adequate for molecular testing: EGFR, KRAS, ALK, HER2, PI3K and BRAF c-MET, and ROS1. They identified 44 mutations, and the most common were KRAS and EGFR mutations [34]. The NAVIGATE study reported the 86.2% adequacy of samples for molecular tests from peripheral tumor lesions when electromagnetic guidance was used rather than radial probe [35]. It was concluded that if a tissue sample of peripherally localized tumors is successfully obtained by any of the bronchoscopically guided techniques, the adequacy of molecular test samples is exceptionally high.

### 2.3. Endobronchial Ultrasound (EBUS) Transbronchial Needle Aspiration

Endobronchial ultrasound-guided transbronchial needle aspiration (EBUS-TBNA) is a minimally invasive bronchoscopic technique with a high diagnostic sensitivity for primarily positive mediastinal lymph node status in patients with LC. This bronchoscopic method is used as a standard method in the diagnostic procedure for LC and its staging. It is performed with a flexible bronchoscope with an incorporated ultrasound guide. This way, a sample can be taken from mediastinal and hilar lymph nodes under direct visualization. This method can provide a sample from the upper and lower paratracheal, prevascular, subcarinal, and hilar lymph nodes (Figure 4). The same technique is used with a gastroscope with an incorporated ultrasound probe, esophageal ultrasound (EUS). With this technique, it is possible to sample the pulmonary ligament, aortopulmonary, subcarinal, and para-esophageal lymph nodes.

Regarding the technical aspects of the EBUS-TBNA method, the CHEST guidelines suggest that to establish the diagnosis of LC with adequate molecular analyses, the samples should be taken in at least three separate penetrations of the needle into the tissue [16] Some authors also suggested the smallest number of three “passes” with the needle in one position of the lymph node if the technique of an immediate analysis of the cytological preparation—rapid on-site ROSE—is applied. If ROSE is not applied, four needle passes per lymph node are required [36]. Other authors investigated whether ROSE increases the diagnostic sensitivity of EBUS-TBNA molecular tests. This randomized study showed the 90.8% specificity of obtaining adequate molecular samples of the tumor tissue by EBUS-TBNA, compared to the 80% specificity in the so-called “non-ROSE” group of patients [37]. Concerning the needle size (21 or 22G), no difference was detected in obtaining an adequate sample for molecular tests, except that there was more blood present in the samples taken with a 19G needle [38,39].

#### RP-EBUS-GS—Guided Endobronchial Ultrasound in the Era of Molecular Testing of Tumor Tissue

A meta-analysis of the adequacy of EBUS-TBNA samples for molecular testing in patients with NSCLC was published. Thirty-three studies including a total of 2698 subjects were analyzed. The pooled probability of obtaining an adequate sample for EGFR and ALK testing was 94.5% and 94.9%, respectively, and the prevalence of EGFR mutation and ALK rearrangement was 15.8% and 2.8%, respectively [40]. A small study from Chile reported that 10 of 12 (83.3%) EBUS-TBNA samples were adequate for ROS1 testing. A retrospective review by Cicek et al. showed that 90 of 98 (91.8%) EBUS-TBNA samples were adequate for ROS1 using the FISH assay [41]. In their paper Xie F et al. documented 100% concordance for EGFR, ALK, and ROS1 results between NGS and conventional analytical assays in EBUS-TBNA samples, with NGS providing information on 12 additional mutations [42].

## 3. PD-L1 Testing on Small Biopsies and Cytology Samples

Current NCCN guidelines recommend quantitative evaluation of tumor cell PD-L1 expression when advanced NSCLC is diagnosed. PD-L1 expression is quantified based on the proportion (%) of viable tumor cell membranes stained for PD-L1 in the total of all viable tumor cells, referred to as the tumor proportion score (TPS). A minimum of 50–100 viable tumor cells is required, depending on the test type. Small biopsies and cytological specimens may increase the risk of false-negative PD-L1 results, based on the established temporal and spatial heterogeneity of the tumor expression [43,44]. More false-negative PD-L1 TPS <1% results were related to smaller samples (<2 mm^2^), while larger samples more often exhibited false-negative for PD-L1 TPS 1–49% results. However, no difference was found in the number of samples detecting PD-L1 TPS ≥50% between small and large biopsies [21,45]. Some authors estimated the concordance of PD-L1 expression between small bronchoscopy samples and surgical resection samples. PD-L1 expression was examined in 79 patients for whom bronchoscopic biopsy and surgical resection specimens were available. The positivity rate of PD-L1 in the transbronchial biopsies samples was 38.0% vs. 35.4% in the resected specimens. Concordance—the congruence of PD-L1 expression between small bronchoscopy and surgical resection samples—was 92.4% [46]. Several recent studies have examined the feasibility of PD-L1 testing from small biopsies and cytology specimens, specifically when examining EBUS-TBNA specimens for PD-L1 IHC analysis. Recent study retrospectively compared the results of PD-L1 testing in EBUS-TBNA samples and in surgical resection specimens in 61 patients, reporting the results of sensitivity, specificity, positive predictive value (PPV), and negative predictive value (NPV) for PD-L1 ≥1% (72%, 100%, 100%, and 80% respectively). The concordance rate between EBUS-TBNA and surgical specimens (cytological vs. histological samples) was 87% for PD-L1 ≥1% and 82% for PD-L1 ≥50%. However, the sensitivity of EBUS-TBNA samples dropped from 72% to only 47% for PD-L1 expression for cutoffs of ≥50% (vs. ≥1%), raising concerns about false-negative PD-L1 results on EBUS-TBNA samples [47]. The available evidence, therefore, suggests that PD-L1 testing of EBUS-TBNA specimens is feasible but also indicates an estimated lower concordance with PD-L1 expression in histological specimens. Future research should therefore define the exact role of PD-L1 expression in cytological bronchoscopy samples for a more accurate prediction of response to immunotherapy.

## 4. Mediastinoscopy and Molecular Testing

Mediastinoscopy, as an invasive diagnostic pulmonological and surgical method, is used mainly to determine the LC stage. A small incision introduces a mediastinoscope through the base of the neck. The sensitivity for determining the presence of tumor cells in mediastinal lymph nodes is 80–95%. False-negative results range from 5–9% due to the impossibility of reaching the para-esophageal, aortopulmonary, and lymphatic nodes of the lower pulmonary ligament with the mediastinoscope. Tissue samples obtained by mediastinoscopy are sufficient for molecular testing. Several studies have compared the diagnostic sensitivity and adequacy results for molecular testing between EBUS-TBNA and mediastinoscopy. As mediastinoscopy provides larger tissue samples, its diagnostic sensitivity was higher; furthermore, mediastinoscopy provides histological samples, unlike EBUS-TBNA, which provides cytological samples, so the adequacy of mediastinoscopy samples for molecular testing is certainly better. The disadvantage of mediastinoscopy is that it is performed under total anesthesia and takes more time [48].

## 5. Transthoracic Needle Aspiration/Biopsy in the Era of Molecular Testing of Lung Tumors

Transthoracic needle aspiration and biopsy (TTNA/TTNB) are safe and quick methods for diagnosing diseases of the lungs, pleura, or mediastinum (Figure 5). However, many clinicians avoid performing these diagnostic methods for peripheral lung lesions because these lesions are potentially resectable; furthermore, if the malignant disease is not proven with TTNA/TTNB, surgery is nevertheless indicated for these patients. These arguments made sense in the early days of cytopathology, when a specific benign diagnosis was rarely achieved. When LC is suspected, a puncture or biopsy of the lesion is necessary to determine the tumor’s cell type and plan further treatment. This is especially true for SCLC, where surgical treatment is not an option. If surgical treatment is contraindicated for any reason, including poor lung function, severe lung emphysema, or a former pneumonectomy, the method is strongly indicated in patients intended for any specific oncological treatment. The indication also exists when a metastatic lung tumor is suspected and in patients with a history of another primary malignancy. TTNA has a diagnostic sensitivity of 80–95% for LC. The sample can be obtained as a needle core biopsy or as a needle aspirate for cytological analysis and cytoblock. Both sample types are suitable for molecular testing as well. Some studies have shown that TTNB samples are adequate for molecular testing, while others have shown that both TTNA and TTNB are equally effective methods for providing tumor tissue samples for molecular analyses [49].

## 6. Thoracocentesis and Thoracoscopy

In the era of molecular therapy and the genotyping and phenotyping of many diseases, malignant pleural effusion (MPE) should not be an exception. MPE is a heterogeneous disease, given its association with different types of malignancy, and future research should address it accordingly. This heterogeneity exists even within the same cancer type. For example, in a study of lung adenocarcinoma with pleural metastases, the mutation EGFR status of the primary tumor did not correlate to that of pleural metastases in 16% of the examined patients [51]. Translational research results obtained in the last decade resulted in a better understanding of MPE pathogenesis, but their clinical application is still poor, and most available data are limited to phase I clinical trials. Intrapleural applications of chemotherapy and immunotherapy have been investigated with contradictory results, and none are yet ready for use. The next decade will certainly bring a better insight into the pathogenesis of MPE and the role of intrapleural chemotherapy or targeted immunotherapy [52]. Thoracoscopy is recommended as a diagnostic method when the thoracocentesis cytological sample is not diagnostically adequate. It is highly sensitive for a malignant pleural disease, unlike the Abrams CT-guided needle pleural biopsy [53]. Thoracoscopy has the sensitivity of 93–97%, a 5 mm sample is usually obtained, and multiple samples can be obtained in a single procedure. The size of these samples is adequate for pathological subtyping and molecular analyses [54]. The tumor biopsy sample should be adequate in size to enable determination of the exact pathohistological subtype of NSCLC and the presence of molecular markers. Based on these findings, specific therapeutic decisions are made. Table 1 displays a comparative review of different invasive pulmonology procedures used to diagnose LC, including the following parameters: the sample type obtained by the respective procedure, diagnostic sensitivity of the sample, and its adequacy for molecular tests [55].

## 7. The Importance of Rebiopsy in the Era of Molecular Therapy for LC

The second or subsequent biopsy—rebiopsy—has become a growing trend in oncology. This is due to more specific/detailed molecular analyses through NGS and new drugs capable of overcoming specific resistance mutations, and a less invasive technique for tissue sampling [56]. Until recently, rebiopsy was performed by a surgeon and intended for malignant tumors, even in advanced stages, and/or reserved for easily accessible sites, such as breast and prostate carcinoma. This concept began to change with the advent of targeted therapy, and numerous new clinical trials that included rebiopsy. For several years, the interest in rebiopsy was confined to clinical research exclusively and excluded from clinical practice. It was not until recently that rebiopsy gained importance in thoracic oncology, with practical implications for both the clinician and the patient. Today, when the liquid biopsy is still to be evaluated, rebiopsy is the only procedure to detect specific transformation and targeted mutations.

### 7.1. Histological Changes in LC

One of the known consequences of resistance to the targeted therapy, especially to EGFR tyrosine kinase inhibitors (TKIs), is a histological transformation within NSCLC. That is the transformation of lung adenocarcinoma (ADC) into SCLC. The frequency of this change is 5% to 14%. The question remains whether SCLC already existed in the tumor tissue itself and manifested itself as a consequence of the applied tyrosine kinase therapy or whether this histological transformation is the result of molecular changes at the level of the RB1 gene, which is the key “trigger” that leads to histological changes in the tumor tissue [57]. Until new studies show the exact mechanism of such histological changes, clinicians should keep such events in mind, and in cases of resistance or rapid disease progression, especially in lung adenocarcinoma treated with TKI, that require rebiopsy.

### 7.2. Acquired Resistance and Rebiopsy in the Era of Molecular Testing

A common concept in both microbiology and oncology is that a prolonged exposure to certain drugs can select a “resistant population”, either a bacterium or a neoplastic cell. In the era of cytotherapy, a common strategy to overcome this phenomenon was to combine different chemotherapeutics with only partial success, limited to hematological malignancies. Acquired resistance has diminished the great hopes placed in targeted therapy. Most NSCLC patients experience, after a variable time interval, the emergence of a resistant clone that causes disease progression. A better understanding of the molecular biology of cancer has, in some cases, enabled the identification of specific mechanisms underlying the acquired resistance occurrence. One of the best examples is T790M in EGFR mutated lung ADC: this mutation is one of the main mechanisms of acquired resistance, generally present only in a minority of cells at diagnosis and much more present in the disease progression. In addition to identifying the mutation, we now have a new tyrosine kinase inhibitor—osimertinib, targeting this type of T790M mutation [58,59,60]. In the coming years, molecular profiling will become a crucial element of patient reevaluation, radiologic evaluations, and physical examinations, making rebiopsy mandatory at disease progression [61].

## 8. Conclusions

In the era of molecular LC therapy, a multidisciplinary approach to diagnosis is strongly recommended to optimize the quality of the obtained tumor tissue and the final therapeutic outcome for patients. Generally speaking, preference should be given to the least invasive diagnostic procedure and obtaining a sample for pathohistological biopsy analysis over cytological analysis. However, there is increasing evidence that, when properly processed, cytology samples can be sufficient for both cancer diagnosis and molecular analysis. A good knowledge of diagnostic procedures is essential for the diagnosis and treatment of LC in the era of personalized therapy.

## Figures and Tables

**Figure 1 medicina-59-01723-f001:**
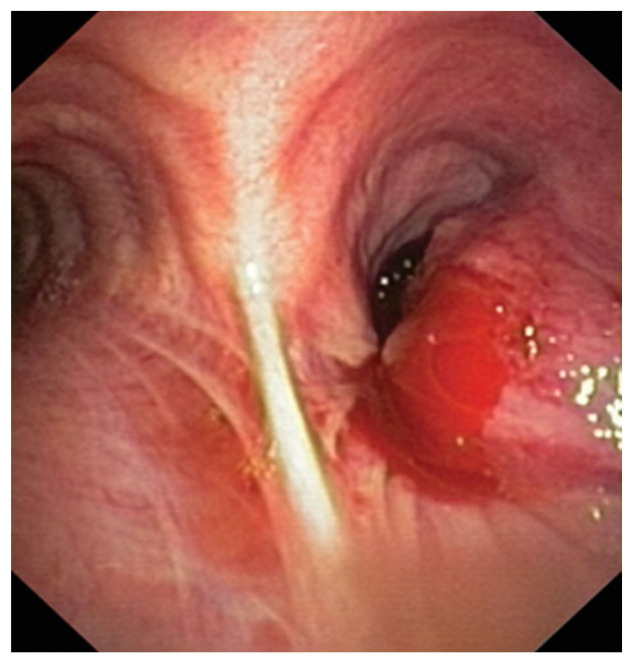
Endoscopy visible tumor in the right main bronchus [17].

**Figure 2 medicina-59-01723-f002:**
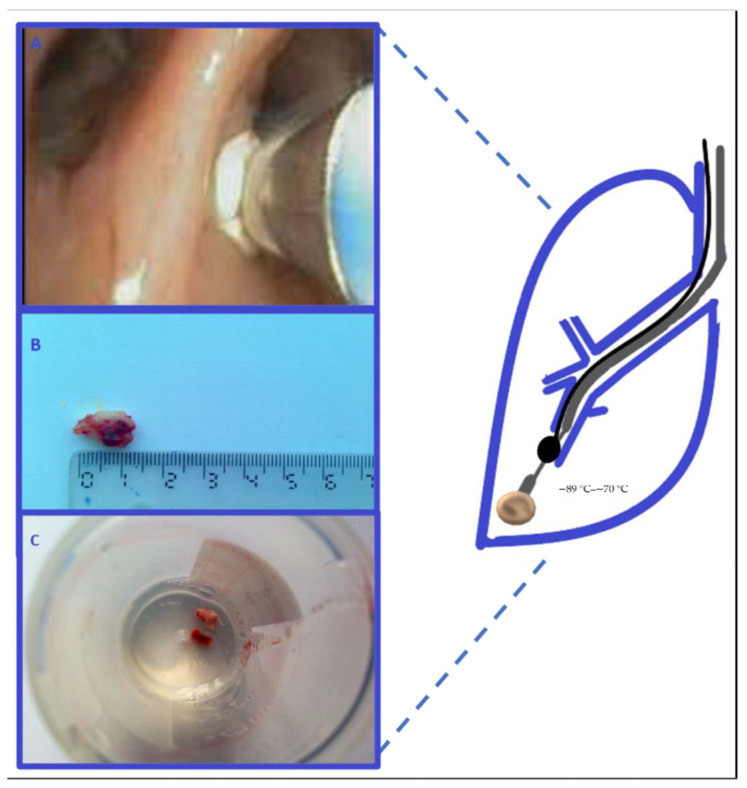
Transbronchial cryobiopsy of the peripheral LC. (**A**)—endoscopic view of the cryoprobe; (**B**,**C**)—macroscopic view of cryobiopsy [23].

**Figure 3 medicina-59-01723-f003:**
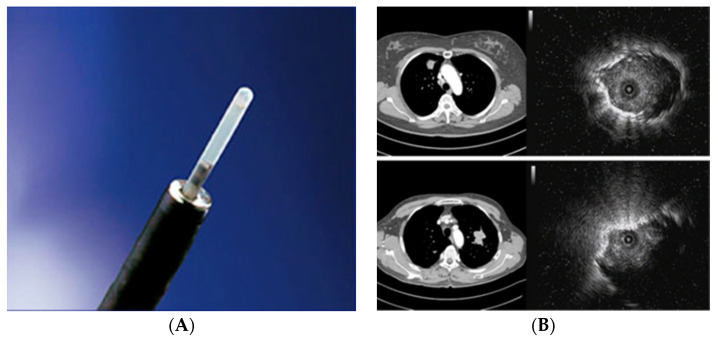
Radial EBUS (**A**). Axial computed tomography scans of the chest and ultrasound images related to the location of the radial probe endobronchial ultrasound (**B**). Upper images—RP within the lesion. Bottom pictures—RP near the lesion [29].

**Figure 4 medicina-59-01723-f004:**
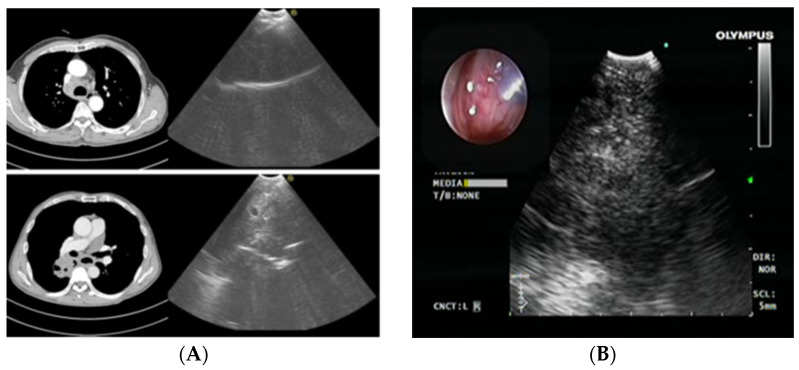
Axial chest computerized tomography (CT) scans (**A**) and a convex probe—EBUS transbronchial needle aspiration (**B**) of the right paratracheal lymph node and the central lung mass [29].

**Figure 5 medicina-59-01723-f005:**
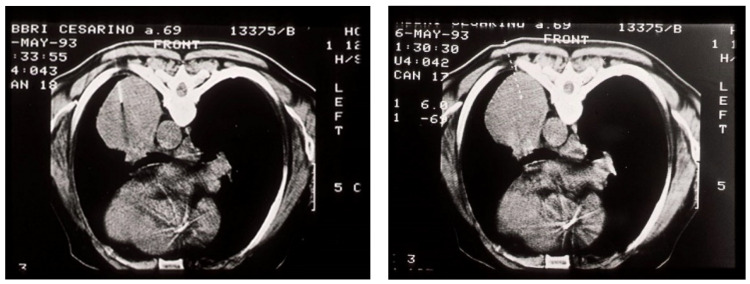
CT image of transthoracic needle puncture (TTNA) of the left lung tumor lesion [50].

**Table 1 medicina-59-01723-t001:** Different invasive pulmonological procedures used to diagnose LC.

Diagnostic Procedure	Sample Type	Diagnostic Sensitivity	Adequacy for Molecular Tests
**Bronchoscopy**	Endobronchial biopsy	70–90% (if the lesion is visible)	100% endobronchial biopsy, less than 50% for lavage
	Brush cytology	The sensitivity increases when combined with bronchial biopsy and lavage	
	Lavage cytology		
**Radial EBUS-GS** (for peripheral lesions larger than 2 cm)	Transbronchial biopsy	58–70% combined with brushing and washing	71% one batch combined with a brush
**Radial EBUS-GS** (for peripheral lesions 2 cm and smaller)	Brush cytology		
	Lavage cytology		
**Mediastinoscopy**	Biopsy	80–95%	It is not set correctly; it depends on the size of the sample
**CT-guided TTNA**	Core needle biopsy	80–95%	100% one batch
	Needle aspiration Cytology		
**Thoracentesis**	Cytology	60–80%	Insufficient rate of 3.7% in one batch
**Thoracoscopy**	Biopsy	93–97%	100% in one batch

Abbreviations: EBUS-GS—endobronchial ultrasound with a guided sheath; TTNA—transthoracic needle aspiration; CT—computed tomography.

## Data Availability

Data are available from authors on reasonable request.

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
