# Peer review of "Invasive Diagnostic Procedures from Bronchoscopy to Surgical Biopsy—Optimization of Non-Small Cell Lung Cancer Samples for Molecular Testing"

_medicina, 2023, doi:10.3390/medicina59101723_

Round 1

Reviewer 1 Report

This study summarized the role and compared the methods of invasive pulmonology diagnostics to obtain adequate tumor tissue samples for molecular analysis, thereby determining the most effective molecular treatments. This study may provide some useful information on the diagnostic procedures from bronchoscopy to surgical biopsy of NSCLC sample for molecular analysis. I have some comments.

1. Page 3, Line 113-118, “It has been proven that five taken samples are the optimal number for reaching the diagnosis~ “.

Please add the references.

2. Page 3, line 128-130, “Several studies have compared the diagnostic sensitivity using cryobiopsy or conventional forceps biopsy in endobronchial lesions. In their work, which included 600 patients in 8 bronchology centers,”

Please add the reference.

3. Page 3, line 130-133, “Hetzel et al. obtained the result that the sensitivity of endobronchial forceps biopsy was 85.1%, and that of cryobiopsies was 95% (p <0.001), thus proving the superiority of cryobiopsy as a diagnostic procedure for patients with LC.”

Please add the reference.

4. Page 3, line 133-135, “Rubio et al. obtained the 96.77%. sensitivity of cryobiopsy for LC. In their study, Kvale et al. documented the diagnostic sensitivity of cryoprobe and conventional forceps biopsy of 92% and 78% respectively [17].

Is this right reference?

(Reference No.17 : Casadevall D, Clavé S, Taus Á , Hardy-Werbin M, Rocha P, Lorenzo M, et al. Heterogeneity of Tumor and Immune Cell PD-L1 Expression and Lymphocyte Counts in Surgical NSCLC Samples. Clin Lung Cancer. 2017;18(6):682-691.e5.)

5. Page 4, line 172-176, “A meta-analysis showed that RP-EBUS has the specificity of 1.00 (95% confidence interval [CI], 0.99–1.00) and the sensitivity of 0.73 (95% CI, 0.70–0.76) in the diagnosis of LC. Sensitivity is higher for lesions larger than 2 cm (77.7%; 95% CI, 73%–82%) than for lesions smaller than 2 cm (56.3%; 95% CI, 51%–61%).”

Please add the references.

6. Page 6, Line 224-227, “Labarca et al. also suggest the smallest number of three "passes" with the needle in one position of the lymph node if the technique of an immediate analysis of the cytological preparation - rapid on-site ROSE is applied. If ROSE is not applied, four needle passes per lymph node are required.”

Please add the reference.

7. Page 6, line 237, “Labarca et al. ~ [37]“.

The first author of this reference is Fernandez-Bussy S. Please modify the sentence to “Fernandez-Bussy S et al.

8. Page 6, line 244, “In their paper, Ksie et al. documented ~ [37]”.

Please correct “Ksie” to “Xie F. et al.”

9. Page 7, line 256, “PDL1 TPS <1%,”

Please correct to “PD-L1”.

10. Page 8, line 305. “small-cell LC”.

Please correct to “SCLC”.

Minor editing of English language required.

Author Response

Dear Sir/Madam,

Thank you for your comments and necessary corrections. We have made the required changes, which we are sending to you in this enclosed letter.

1. Page 3, Line 113-118, “It has been proven that five taken samples are the optimal number for reaching the diagnosis~ “. Please add the reference.

We add the reference:

Wahidi MM, Herth F, Yasufuku K, et al. Technical Aspects of Endobronchial Ultrasound-Guided Transbronchial Needle Aspiration: CHEST Guideline and Expert Panel Report. Chest 2016;149:816-35.

2.  Page 3, line 128-130, “Several studies have compared the diagnostic sensitivity using cryobiopsy or conventional forceps biopsy in endobronchial lesions. In their work, which included 600 patients in 8 bronchology centers,” Please add the reference.

We add the reference:

Arimura, K.; Tagaya, E.; Akagawa, H.; Nagashima, Y.; Shimizu, S.; Atsumi, Y.; Sato, A.; Kanzaki, M.; Kondo, M.; Takeyama, K.; et al. Cryobiopsy with endobronchial ultrasonography using a guide sheath for peripheral pulmonary lesions and DNA analysis by next generation sequencing and rapid on-site evaluation. Respir. Investig. 2019, 57, 150–156.

3. Page 3, line 130-133, “Hetzel et al. obtained the result that the sensitivity of endobronchial forceps biopsy was 85.1%, and that of cryobiopsies was 95% (p <0.001), thus proving the superiority of cryobiopsy as a diagnostic procedure for patients with LC.” Please add the reference.

We add the reference:

Hetzel, R. Eberhardt, F.J.F. Herth, et al. Cryobiopsy increases the diagnostic yield of endobronchial biopsy: a multicentre trial. Eur Respir J 2012; 39: 685–690.

4. Page 3, line 133-135, “Rubio et al. obtained the 96.77%. sensitivity of cryobiopsy for LC. In their study, Kvale et al. documented the diagnostic sensitivity of cryoprobe and conventional forceps biopsy of 92% and 78% respectively [17]. Is this right reference?

(Reference No.17 : Casadevall D, Clavé S, Taus Á , Hardy-Werbin M, Rocha P, Lorenzo M, et al. Heterogeneity of Tumor and Immune Cell PD-L1 Expression and Lymphocyte Counts in Surgical NSCLC Samples. Clin Lung Cancer. 2017;18(6):682-691.e5.)

We correct the reference:

Rubio, E.R.; Le, S.R.; Whatley, R.E.; Boyd, M.B. Cryobiopsy: Should This Be Used in Place of Endobronchial Forceps Bi-opsies? BioMed Res. Int. 2013, 2013, 730574.

5. Page 4, line 172-176, “A meta-analysis showed that RP-EBUS has the specificity of 1.00 (95% confidence interval [CI], 0.99–1.00) and the sensitivity of 0.73 (95% CI, 0.70–0.76) in the diagnosis of LC. Sensitivity is higher for lesions larger than 2 cm (77.7%; 95% CI, 73%–82%) than for lesions smaller than 2 cm (56.3%; 95% CI, 51%–61%).” Please add the references.

We add the reference:

Ahn JH. An update on the role of bronchoscopy in the diagnosis of pulmonary disease. Yeungnam Univ J Med [Internet]. 2020 Oct 31;37(4):253–61. doi: 10.1016/j.resinv.2018.10.006. [00-26].

6. Page 6, Line 224-227, “Labarca et al. also suggest the smallest number of three "passes" with the needle in one position of the lymph node if the technique of an immediate analysis of the cytological preparation - rapid on-site ROSE is applied. If ROSE is not applied, four needle passes per lymph node are required.” Please add the reference.

We add the reference:

Labarca, E. Folch, M. Jantz, H.J. Mehta, A. Majid, S. Fernandez-Bussy.Adequacy of samples obtained by endobron-chial ultrasound with transbronchial needle aspiration for molecular analysis in patients with non-small cell lung can-cer. Systematic review and meta-analysis. Ann Am Thorac Soc, 15 (2018), pp. 1205-1216.

7. Page 6, line 237, “Labarca et al. ~ [37]“.

The first author of this reference is Fernandez-Bussy S. Please modify the sentence to “Fernandez-Bussy S et al.

We modify the sentence into:

Fernandez et al. published a meta-analysis of the adequacy of EBUS-TBNA samples for molecular testing in patients with NSCLC.

8. Page 6, line 244, “In their paper, Ksie et al. documented ~ [37]”.

Please correct “Ksie” to “Xie F. et al.”

We correct the name into the sentence:

In their paper Xie F et al. documented 100% concordance for EGFR, ALK, and ROS1 re-sults between NGS and conventional analytical assays in EBUS-TBNA samples, with NGS providing information on 12 additional mutations [39].

9. Page 7, line 256, “PDL1 TPS <1%,”

Please correct to “PD-L1”.

We corrected the required:

More false-negative PD-L1 TPS <1%, results were related to smaller samples (<2 mm2), while larger samples more often exhibited false-negative for PD-L1 TPS 1–49% results.

 10.  Page 8, line 305. “small-cell LC”.

Please correct to “SCLC”.

We corrected the required:

When LC is suspected, a puncture or biopsy of the lesion is necessary to determine the tumor's cell type and plan further treatment. This is especially true for SCLC where a sur-gical treatment is not an option.

The corrected version of the paper is attached.

Reviewer 2 Report

The paper aims to conduct an exhaustive investigation into the efficacy of invasive diagnostic techniques employed in lung cancer diagnosis, with a specific emphasis on optimizing tissue sampling for molecular testing and underlining the significance of rebiopsy in the context of molecular therapy. Its primary contributions include a thorough examination of various diagnostic approaches, a strong focus on molecular testing relevance, and the acknowledgment of the evolving role of rebiopsy. Strengths of the paper encompass a well-organized structure and a clear commitment to scientific content. Nevertheless, there are certain notable areas of weakness. For instance, the article lacks a comprehensive discussion of potential complications associated with invasive procedures, such as pneumothorax, which could significantly impact patient outcomes. Additionally, the review might benefit from a more detailed cost-effectiveness analysis of the different diagnostic methods discussed. To enhance its scientific rigor and practical applicability, it is recommended that the authors address these weaknesses and consider the incorporation of these suggestions.

The paper presents a compelling argument regarding the importance of optimizing invasive diagnostic procedures in lung cancer diagnosis. It thoroughly reviews various techniques, from bronchoscopy to transthoracic biopsy, providing readers with a broad understanding of available options. The focus on molecular testing and its relevance in the era of targeted therapies is commendable and aligns with current trends in oncology. The discussion around the need for rebiopsy in the context of disease progression and therapy resistance is also pertinent. However, the paper could benefit from a more critical examination of the potential complications and risks associated with these invasive procedures. For instance, when discussing transthoracic biopsies, it would be valuable to elaborate on the incidence of pneumothorax and its management. Moreover, while the paper touches on cost considerations briefly, a more in-depth cost-effectiveness analysis comparing these diagnostic methods could add substantial value for clinicians and decision-makers.

The review's completeness and relevance to the field of lung cancer diagnosis and treatment are noteworthy. It effectively identifies the knowledge gap regarding the optimization of tissue sampling and the role of rebiopsy in personalized therapy decisions. The selection of references is generally appropriate, and the paper draws upon a range of credible sources to support its arguments. However, to further strengthen the paper, it would be beneficial to include more recent studies and clinical trials, as the field of lung cancer diagnostics is rapidly evolving. Additionally, while the review primarily focuses on the benefits and limitations of each diagnostic technique, it could be enhanced by providing practical guidelines or recommendations for clinicians regarding the selection of the most suitable approach based on specific patient profiles.

Author Response

Dear Sir/Madam,

Thanks for the comprehensive commentary on the entire paper. Thank you for the observed necessary corrections too. We have made the explanations of the same, which we are sending to you in this enclosed letter:

As for the potential complications of the mentioned invasive procedures, pneumothorax is a rare complication of all the mentioned procedures, except transthoracic needle puncture, which is treated with drainage only if necessary. The purpose of this paper was to compare the adequacy of samples for molecular testing of tumors by comparing different techniques of invasive pulmonological procedures, including surgical ones. The idea of ​​the paper was not to compare the techniques of invasive pulmonology diagnostics and the complications of the mentioned techniques, because that would change the structure of the paper and move away from the given topic. Cost-effectiveness analysis comparing these diagnostic methods could not be compared because we did not have such information, therefore comparisons of the results related to the given topic of many authors from different countries could not be shown.

In the previous paragraph, we explained everything about the comments about the potential complications of the presented procedures and the cost-effectiveness. because it seems to us that you mentioned that query twice.

About the selection of the most suitable approach to invasive pulmonology diagnostics, as you mentioned (like some guidelines or recommendations for clinicians), we have shown the best results in terms of the suitability or sensitivity of the taken tumor samples for all molecular testing, and have shown the eventual need for repeated sampling (rebiopsy or perform the liquid biopsy) according to changes in the molecular structure of the tumor during lung cancer treatment, which are not consequences of specific patient profiles. We listed a lot of recent studies and clinical trials about the topic.

Reviewer 3 Report

The authors have generalized the useful techniques of biopsy in the diagnosis of lung cancer, encompassing traditional bronchoscopy, cryobiopsy, EBUS-TBNA, EBUS-GS, CT-guided biopsy, mediastinoscopy, and others. This is a vast topic that cannot be adequately summarized in a brief review without introducing novel insights into our knowledge.

A structured introduction, which includes indications, contraindications, merits, demerits, key examination points, diagnostic accuracy, and more, would prove to be more valuable for clinical practice.

Author Response

Dear Sir/Madam, Thanks for the comprehensive commentary on the entire paper. Thank you for the observed necessary corrections too. We have made the explanations of the same, which we are sending to you in this enclosed letter:

The topic of the review paper was not a comparison of invasive pulmonology procedures in the diagnosis of lung cancer, but a comparison of all "old" as well as new invasive diagnostic procedures in obtaining adequate samples of tumor tissue for molecular biological tests, which is still the subject of comparison and examination with the expectation of even more relevant results.

The introduction summarizes all diagnostic procedures and their use in the era of molecular lung tumor testing. The entire paper provides accurate comparisons of all procedures in terms of obtaining the most adequate samples for molecular diagnostics as a condition for the application of the latest molecular therapies for lung cancer. comparison of invasive pulmonology procedures including indications, and contraindications was not the topic of our review paper.

Round 2

Reviewer 3 Report

I have no additional comment on this manuscript.